# The Greek Burnout Assessment Tool: Examining Its Adaptation and Validity

**DOI:** 10.3390/ijerph20105827

**Published:** 2023-05-15

**Authors:** George S. Androulakis, Dimitra Ap. Georgiou, Olga Lainidi, Anthony Montgomery, Wilmar B. Schaufeli

**Affiliations:** 1Department of Business, Administration University of Patras, 26504 Patras, Greece; gandroul@upatras.gr; 2Department of Educational Sciences and Social Work, University of Patras, 26504 Patras, Greece; dgeorgioy@upatras.gr; 3School of Psychology, University of Leeds, Leeds LS29JT, UK; o.lainidi@leeds.ac.uk; 4Department of Psychology, Northumbria University, Newcastle NE18ST, UK; 5Research Group Work Organizational and Personnel Psychology (WOPP), 3000 Leuven, Belgium; wilmar.schaufeli@kuleuven.bes; 6Department of Social and Organizational Psychology, Utrecht University, 3584 CS Utrecht, The Netherlands

**Keywords:** burnout, Greek adaptation, Rasch analysis, reliability, validity

## Abstract

Burnout is a significant challenge in the workplace. Its extent is global and its unfavourable consequences are diverse, affecting the individual, the organization, and society. The aim of the present study was to examine the adaptation and assess the validity of the Greek version of the Burnout Assessment Tool (BAT). The adaptation process included the translation and back-translation of the BAT. Data were collected from 356 Greek employees from diverse sectors. Confirmatory factor analysis and item response theory were utilized to assess the validity of the Greek version of the BAT. According to the findings of the present research, the core symptoms scale and the secondary symptoms scale of BAT-23 and BAT-12 models demonstrated adequate structures for the analysis and measurement of burnout in the Greek context. Finally, the psychometric performance of the BAT-GR-12 compared to the BAT-GR-23 establishes it as a more optimum instrument for the assessment of burnout across Greek working adults.

## 1. Introduction

According to the World Health Organization, burnout is an occupational syndrome resulting from chronic stress within the workplace that has not been managed effectively and is characterized by three elements: feelings of physical and emotional exhaustion or energy depletion, feelings of cynicism or detachment towards one’s work, and a reduced sense of professional efficacy [1].

The dimensions of exhaustion and cynicism are viewed as essential to burnout [2,3,4], whereas personal accomplishment also contributes to the syndrome without always being considered a structural element of burnout [5,6,7]. Various tools exist for measuring and assessing burnout, such as the Maslach Burnout Inventory [8], the Oldenburg Burnout Inventory [9], the Bergen Burnout Inventory [10], the Copenhagen Psychosocial Questionnaire [11], the Spanish Burnout Inventory [12], the Granada Burnout Questionnaire [13], the Burn Out-Neuratshenia Complaints Scale [14], the Shirom Melamed Burnout Measure [15], the Burnout Measure [16], and the Boudreau Burnout Questionnaire [17].

Recent research findings have established that burnout is also characterized by reduced cognitive performance [18], cognitive deficits [19], and diminished visuospatial abilities [20]. This has given rise to the development of a new approach to assess burnout; the Burnout Assessment Tool (BAT) [21,22]. The BAT (unlike the MBI) includes a dimension which measures cognitive impairment. Moreover, the overemphasis on the emotional exhaustion dimension as being the core component of burnout (e.g., [23]) has resulted in (a) an increased number of published research that equates burnout primarily with emotional exhaustion (EE) [24], (b) leading to other dimensions being considered less important; and (c) significantly restricting the potential of other dimensions as important to furthering our understanding of burnout. Thus, the BAT was developed in line with the theoretical description of burnout where the focus is on both the inability and unwillingness to put effort [25], as it includes both a dimension of exhaustion and mental distancing, respectively. The BAT questionnaire was designed as a diagnostic tool and at the same time as a potential screening tool because it allows healthy workers to be distinguished from those at risk of burnout [22].

The BAT questionnaire measures the central dimensions of burnout as well as the secondary dimensions of the syndrome (33 questions in total). The self-report questionnaire contains four (4) core dimensions, three (3) of which refer to the inability to invest energy (exhaustion, cognitive and emotional impairment), and one (1) refers to the unwillingness to invest the required energy (mental distance). In addition, there are three secondary dimensions to the questionnaire that usually coexist with the main symptoms which are depressed mood, psychological distress, and psychosomatic complaints. Additionally, the developers of the BAT also created a short version which includes 12 questions (BAT-12) [21]. The psychometric properties of the BAT have already been evaluated in countries such as Austria, Belgium, Brazil, Ecuador, Finland, Germany, Ireland, Italy, Japan, Korea, Netherlands, New Zealand, Poland, Portugal, Romania, Russia, Sweden, and Turkey [22]. The results of the research by De Beer et al. (2020) [26] on the comparison of the effect sizes of the latent means of BAT between six European countries (The Netherlands, Belgium, Germany, Austria, Ireland, and Finland) and Japan showed that Japan had a significantly higher score on overall burnout as well as on all first-order factors. No significant differences were observed between the European countries in the overall burnout scores, while only some minor differences were noted in the first-order factors between some of the countries. Based on the exploratory factor analysis (EFA) of the BAT manual [22], the four core factors can be conceived as the four fundamental dimensions. Notably, the dimension of exhaustion accounts for a significantly higher amount of variance than the other three core factors. Similar results have emerged in Italian samples [27,28]. In Poland, in a sample of workers, the four dimensions of the BAT were confirmed using a bifactor model [29]. In Korea, the CFA confirmed the structure of the four main factors, although the EFA indicated the exclusion of item ex06 [30]. Validity evidence for both BAT-23 and BAT-12 was obtained from data in Brazil and Portugal, using the item response theory and CFA in conjunction with the classical test theory [31]. Moreover, the Ecuadorian version of the BAT indicated that the hierarchical model for both BAT-23 and BAT-12 was suitable for the Ecuadorian context [32]. For the Turkish version of BAT evidence supported the six-factor structure with four primary and two secondary factors identified [33].

The present study assessed the validity and reliability of the BAT (named the BAT-GR hereafter) for use in the Greek context. Burnout has most frequently been studied in relation to both work engagement [34] and anxiety/depression [35]. Work engagement and burnout are considered to be two opposite constructs [4,35], with research indicating that burnout and engagement are indeed interconnected but represent different forms of employee well-being [4,36]. Congruently, anxiety/depression have been hypothesized to be positively related to burnout, with evidence suggesting that there is an overlap between burnout and depression [35]. Therefore, work engagement and anxiety/depression were measured to assess the discriminant validity of the BAT-GR.

Following the Standards for Educational and Psychological Testing proposals [37], this work aimed at exploring the validity of the BAT regarding both the internal structure, and the relations with the variables of work engagement and anxiety/depression. The research undertaken aimed to (a) translate the BAT for use in Greek; (b) examine the reliability and factorial validity of the translated BAT-GR; and (c) examine the discriminant validity vis-a-vis work engagement, depression, and anxiety.

## 2. Methodology

### 2.1. Participants

In order to explore the psychometric properties of the BAT-GR, data were collected from a workforce representative sample belonging to various economic sectors. A total of 2500 workers, randomly selected, received a link to anonymously fill out an online questionnaire; 629 of them responded and finally, 356 of them completed all the questions of the questionnaire, thus resulting in a response rate of 0.14. The descriptive statistics of the sample are provided as Appendix A, “SupMat1_Descreptive_statistics_sample” [38]. The sample is representative of the working population in Greece (the percentages of employees in the sample do not differ significantly from the corresponding percentages of employees per economic sector in Greece) [39].

### 2.2. Data Collection

The survey was conducted between September 2021 and March 2022. All participants answered the questionnaire via a web-designed platform. Participation was voluntary and the participants were required to agree to a Consent Form prior to filling out the survey. Ethical approval for the research was approved by the Research Ethics Committee of the University of Patras (No. 14511).

### 2.3. Measures and Instruments

The Burnout Assessment Tool measures burnout encompassing the core dimensions (also core symptoms, C.S.) of exhaustion (Exh), 8 items; mental distance (M.D), 5 items; cognitive impairment (C.I), 5 items; and emotional impairment (E.I), 5 items, as well as the secondary dimensions (also secondary symptoms, S.S.) of psychological distress (Psychl.C), psychosomatic complaints (Psych.C) and depressed mood (10 items). The total BAT questionnaire consists of 33 items, where 23 assess the 4 core dimensions (named BAT-23), while the other 10 items compose the secondary dimensions. In addition, BAT-12 consists of twelve items from BAT-23, three from each core dimension.

The Utrecht Work Engagement Scale (UWES-17) was employed to assess work engagement via 17 items and a 7-point scale from 0 = “never” to 6 = “daily.” UWES items are categorized into 3 subscales expressing the 3 dimensions of work engagement; vigor (VI), 6 items; dedication (DE), 5 items; and absorption (AB), 6 items (Schaufeli and Bakker, 2004). High scores on dedication, vigor, and absorption indicate a high level of engagement.

The Hospital Anxiety and Depression Scale (HADS) was used to evaluate states of anxiety (A) and depression (D) via two 7-item subscales measuring each of the above states [40,41,42].

### 2.4. Procedures

#### Translation and Adaptation

The English-to-Greek translation of the initial BAT questionnaire, in terms of both the core and the secondary burnout symptoms, was conducted independently by two certified experts [43]. The synthesis of the two translated versions was accomplished by the first four authors of the paper (Greek version can be found at https://burnoutassessmenttool.be/wp-content/uploads/2021/08/BAT-Greek.pdf accessed on 18 April 2023). Following this, another bilingual (Greek and English) expert, who had not read the original items, conducted back-translation from Greek into the English language. The original English and the back-translated versions were compared and harmonized and also approved by the authors of the original instrument.

### 2.5. Data Analysis

A confirmatory factor analysis (CFA) using maximum likelihood estimation was applied to examine the factorial validity of BAT-GR-23 and BAT-GR-12. The fit measures were evaluated for all models. BAT-GR-23 and BAT-GR-12 were assessed via a unifactorial structure where all items represented a uniform score for burnout. Figure 1 illustrates the second-order structure of BAT-GR-23 and Figure 2 of BAT-GR-12, accordingly. Additionally, a third-order model was tested expressing the sum of the core symptoms of the subscales of BAT-GR-23 along with the secondary factors, presented in Figure 3. Goodness of fit for each model was assessed via the indices suggested by Kline (2015) [44]: Chi-squared (chisq), degrees of freedom (df), *p*-value (P) of 2 df ratio, the comparative fit index (CFI), the Tucker–Lewis index (TLI), the root mean square error of approximation (RMSEA), the confidence interval of RMSEA (lower and upper value) and the corresponding *p*-value (pv). An acceptable fit for CFI and TLI occurs when the values are higher than 0.90—and preferably higher than 0.95—and for RMSEA occurs when they are equal or less than 0.08 [45]. BAT-GR-23 and BAT-GR-12 scales were examined employing Cronbach’s alpha, and values higher than 0.70 are considered adequate while those higher than 0.80 are considered good [46]. BAT-GR-23 and BAT-GR-12 were analyzed in terms of convergent validity via the intercorrelations between BAT-GR and each structural element of work engagement and anxiety/depression. The statistical package R was used for data processing [47].

## 3. Results

The basic statistics (mean, standard deviation, skewness, and kurtosis) and the frequencies (never, rarely, sometimes, often, and always) for all items of both the core and secondary dimensions of BAT-GR are provided as Appendix A, “SupMat2_Descreptive_statistics_core” and “SupMat3_Descreptive_statistics_secondary” [38]. It can be observed that the level of burnout in the sample examined is at the threshold to be characterized as high on the basis of the comparison with data from the BAT manual [21,22]. Note that, for both BAT-GR-23 and BAT-GR-12, no statistically significant differences were found in terms of gender (*p*-value = 0.93 and 0.50, accordingly), age (*p*-value = 0.22 and 0.82, accordingly), and level of education (*p*-value = 0.11 and 0.31, accordingly).

### 3.1. Factorial Validity

Table 1 includes the fit measures concerning the confirmatory factor analysis of all models mentioned in this paper and the Satorra–Bentler x^2^ comparisons between nested models [48]. Specifically, in Table 1 the models indicated are BAT-GR-23 and BAT-GR-12 second-order structures, bi-BAT-GR_23 and bi-BAT-GR-12 bifactor models, and the second-order secondary symptoms structure. Ιn Table 1, the label (A) indicates ‘Adequate fit and the label (G) indicates Good fit. The CFI, TLI, SRMR, and RMSEA indices of BAT-GR-12 are slightly better than those of BAT-GR-23, and therefore the shorter scale fits the data significantly better than the original. Note that, only for BAT-GR-23 the CFI and TLI indices are slightly below the threshold of 0.90 while SRMR and RMSEA indices are adequate and less than 0.08. For BAT-GR-12 and the bifactor models all the indices are adequate. The indices for bifactor models are better than those for both BAT-GR-23 and BAT-GR-12.

Figure 4 and Figure 5 illustrate accordingly the burnout structural bifactor models bi-BAT-GR-23 and bi-BAT-GR-12. For bi-BAT-GR-23 all loadings are statistically significant, having a *p*-value lower or equal to 0.05. It appears that all loadings are positive, except for e06 which shows a different sign between first and second-order loadings. Note that this could occur in items with high discrimination ability as can be seen in the next subsection. For bi-BAT-GR-12, all loadings are statistically significant, having a *p*-value lower than 0.05. It appears that all loadings are positive, except for md13 which is negative but weak. Additionally, for the evaluation of bifactor models, the hierarchical omega index was used [49,50]. The hierarchical omega index for the GR-BAT-23 with four factors was calculated to be ω_h_ = 0.73, while for the GR-BAT-12 it was found to be ω_h_ = 0.71, which in both cases is greater than 0.70, thus supporting the tenability of these factors.

### 3.2. Reliability

The internal consistency indices are illustrated in Table 2. For both BAT-GR-23 and BAT-GR-12, the reliability indicators are acceptable. Focusing on their subscales, all reliability indicators ranged between 0.70 and 0.85, and the only indicator that appears on the lower acceptable “threshold” concerns the subscale mental distance of BAT-GR-12 (alpha = 0.67).

Rasch analysis is widely used in order to examine the reliability of a scale as it is a strong test of homogeneity and provides information about the soundness of each item [51,52,53,54]. The information regarding the item information function of the Rasch Model of all BAT-GR items for both the core and secondary symptoms is provided as Appendix A, “SupMat4_Rasch_analysis” [38]. For exhaustion, the item ex06 displays the greatest discrimination ability of the items in BAT-GR-23 but not in BAT-GR-12, which also explains the fact why the correlation coefficient of BAT-GR-23 with BAT-GR-23-Exh was higher than that of BAT-GR-12-Exh with BAT-GR-12. In addition, the item information curves are grouped homogeneously which is a positive outcome of the Rasch analysis. Moreover, it is illustrated that the curves’ peaks are high, exceeding 0.6, indicating the items’ ability to accurately measure the dimensions. Considering the above, the BAT-GR items can be adequately adapted and applied to the Greek context for the measurement of burnout.

### 3.3. Construct Validity and Relation with External Variables

In order to assess the discriminant validity of the BAT-GR, all dimensions were assessed in relation to work engagement and anxiety/depression.

Cronbach’s alpha for UWES was 0.94 and for HADS is 0.85, which are greater than 0.7. For the correlation coefficients between BAT-GR-23, UWES-17 and HADS as well as their individual subscales, burnout and UWES were negatively correlated (r = −0.45). No statistically significant correlation was found between the subscale absorption with the secondary symptoms subscales of burnout. For the HADS components, there was a positive correlation with BAT-GR-23 (0.56 for both anxiety and depression), and a negative correlation with engagement, anxiety (−0.16), and depression (−0.38). In addition, anxiety was correlated with emotional impairment (0.55) and exhaustion (0.52), while depression was correlated with mental distance (0.53) and exhaustion (0.51). Furthermore, anxiety showed a negative correlation with vigor (−0.24) while depression was negatively correlated with both dedication (−0.40) and vigor (−0.39). Similar results were reported for the BAT-GR-12, in relation to both UWES and HADS, suggesting that both scales are suitable for use in the Greek context.

In order to investigate the convergent and discriminant validity of the BAT-GR in relation to work engagement and HADS, the Average Variance Explained (AVE) and square latent correlations R^2^ work engagement (UWES), HADS and BAT-GR were conducted and the results are presented in Table 3. AVE exceeded the square correlation R^2^ of both the HADS and BAT-GR latent factors, which further supports the discriminant validity of the BAT-GR.

## 4. Discussion and Conclusions

The aim of the present study was the assessment of the Burnout Assessment Tool [21,22] for use in the Greek context. The core dimensions of the BAT-GR- exhaustion, mental distance, and emotional and cognitive impairment—were robustly supported by the results of the factorial loadings, the reliability of the dimensions as well as the goodness-of-fit indices [22,55].

The findings of the present work indicate that both the BAT-GR-23 and BAT-GR-12 can be used across the Greek working population to examine an extensive level of burnout manifestations, as well as to measure adequately all the core and secondary dimensions of the syndrome, that is, exhaustion, mental distance, emotional impairment, cognitive impairment, psychological and psychosomatic complaints.

Alpha, Omega, and composite reliability indices of all BAT-GR dimensions, with the exception of mental distance from BAT-GR-12, were acceptable. Rasch analysis also supported the homogeneity and robustness of the items permitting and therefore supporting the application of the instrument for the evaluation of burnout along with its core and secondary dimensions. For both BAT-GR-23 and BAT-GR-12, bifactor models were applied in order to advance the precision of psychometric measurements and enhance the reliability of the results, revealing the existence of the general burnout factor while ensuring that the correlations between the core factors are accurately interpreted.

Concerning the interrelations between burnout and work engagement, the present work indicates that these two constructs constitute different and opposing work-related phenomena. In particular, convergent validity of both BAT-GR-23 and BAT-GR-12 was analyzed through the intercorrelations developed between BAT-GR with each structural element of work engagement. The positive relations between burnout and its core and secondary dimensions were supported, as well as the negative relationships between burnout and work engagement. In terms of the correlations between burnout, HADS, anxiety, and depression, the present work reveals that these constructs constitute different phenomena, as shown via the positive correlations between them. Moreover, discriminant validity is supported as the AVE of the BAT-GR is greater than the squared correlation (R^2^) of UWES and HADS. The above results, combined with the factorial validity conducted via CFA and bifactor analysis, add further evidence to the robustness of the BAT-GR. This means that burnout can be considered a syndrome that consists of different symptoms (i.e., dimensions) that refer to one underlying construct (i.e., burnout).

The examination of the studied variables relied solely on self-report questionnaires which includes the risk of the common method variance [55]. Chance capitalization is always an issue when one examines multiple associations. Thus, it is possible that statistically significant results occur due to chance. The sample size was adequate and representative of the working population, but not in terms of gender and age. This may be another limitation of the present work, although comparisons (t-tests) of gender and age groups did not reveal statistically significant differences. A further limitation that could be considered is that the same sample was used for both GR-BAT-23 and GR-BAT-12. An additional limitation may be the low response rate of 14%. This could be explained by the fact that the data collection occurred in a period where the working conditions were adjusted according to the restrictions of the pandemic COVID-19 and consequently the corresponding answers could have been influenced by these circumstances.

Considering all the above, the present study provides evidence that the all BAT-GR items can be adequately adapted and applied to the Greek context for the measurement of burnout. The psychometric performance of the BAT-GR-12 compared with the BAT-GR-23 shows that the former is a more suitable instrument for the assessment of burnout based on the goodness-of-fit indices of the second-order schema. Nevertheless, the present research provides initial evidence for the psychometric acceptability and applicability of the BAT-GR within the Greek context.

## Figures and Tables

**Figure 1 ijerph-20-05827-f001:**
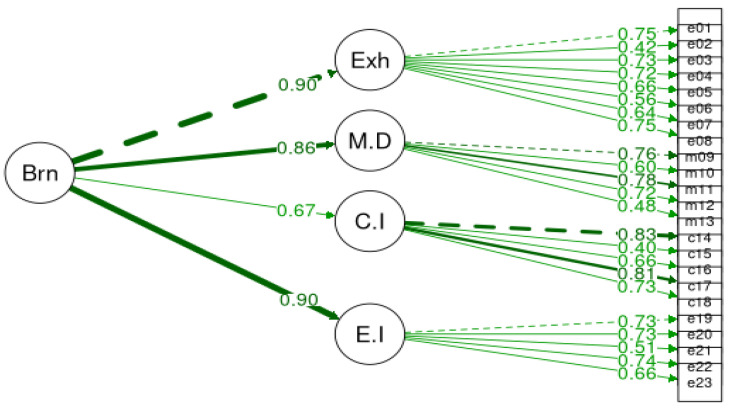
Burnout structural factor model for BAT-23.

**Figure 2 ijerph-20-05827-f002:**
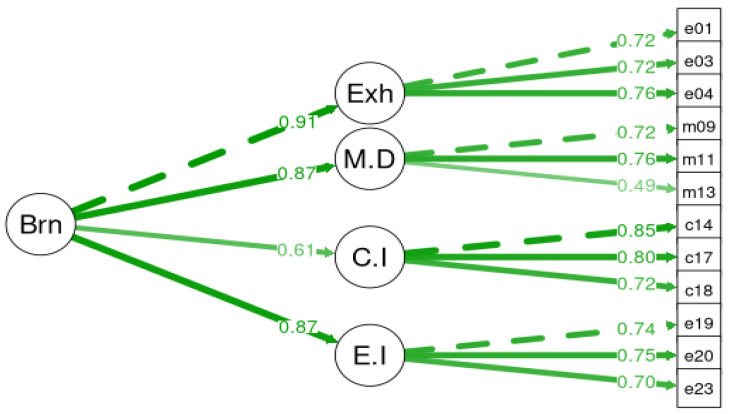
Burnout structural factor model for BAT-12.

**Figure 3 ijerph-20-05827-f003:**
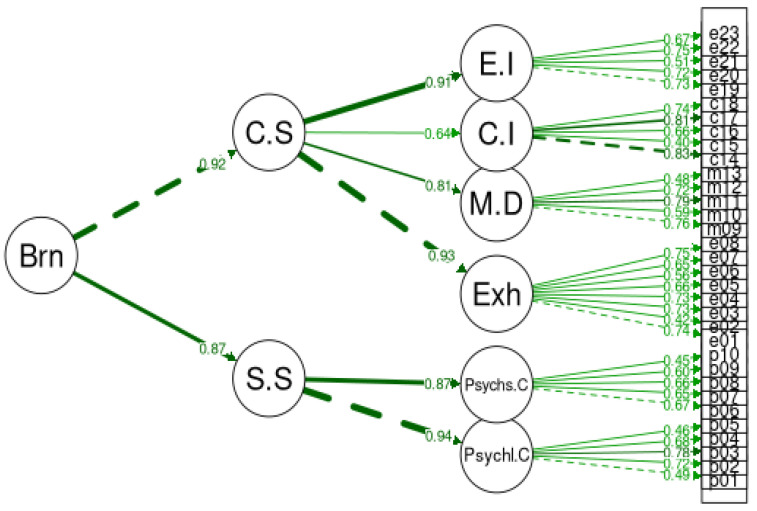
Burnout structural factor model for BAT.

**Figure 4 ijerph-20-05827-f004:**
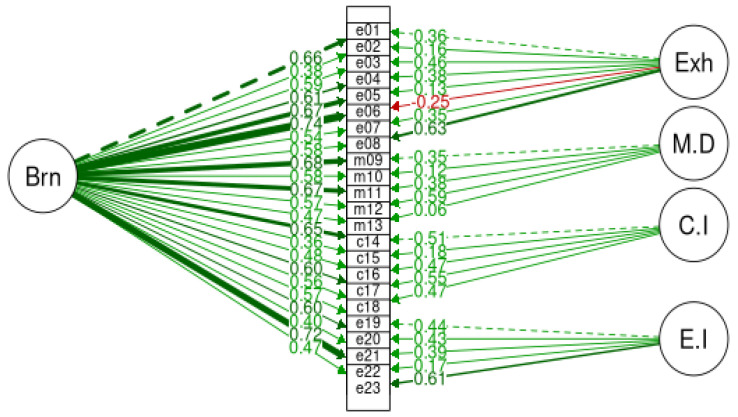
Burnout structural factor model for bi-BAT-23.

**Figure 5 ijerph-20-05827-f005:**
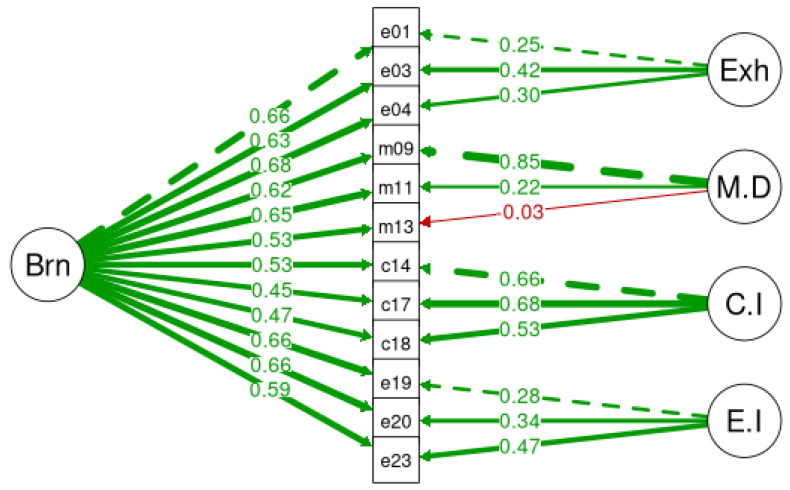
Burnout structural bifactor model for BAT-12.

**Table 1 ijerph-20-05827-t001:** CFA fit measures and alternative model comparisons.

Models	chisq	df	*p*	CFI	TLI	SRMR	RMSEA	Lower	Upper
BAT-GR-23(A)	632.7226	226	0.000	0.8882	0.8748	0.0736	0.0713	0.0648	0.0779
bi-BAT-GR-23(G)	409.0899	207	0.000	0.9444	0.9321	0.0564	0.0525	0.0450	0.0600
BAT-GR-12(G)	125.7871	50	0.000	0.9557	0.9415	0.0537	0.0654	0.0513	0.0798
bi-BAT-GR-12(G)	102.1259	42	0.000	0.9648	0.9447	0.0469	0.0636	0.0480	0.0794
Secondary Symptoms (G)	59.6993	33	0.003	0.9724	0.9623	0.0372	0.0477	0.0275	0.0667
Second-order BAT-GR (A)	1103.9040	487	0.000	0.8761	0.8657	0.0726	0.0598	0.0551	0.0645
**Models comparison**	**Δchisq**	** *p* **		**df**	**ΔCFI**	**ΔTLI**	**ΔSRMR**		**ΔRMSEA**
bi-BAT-GR-23–BAT-GR-23	223.6300	0.0000		19	0.0562	0.0573	−0.0172		−0.0211
bi-BAT-GR-12–BAT-GR-12	23.661	0.0026		8	0.0091	0.0032	−0.0018		−0.0033

chisq = Chi-square; df = degree of freedom; P, *p*-value; CFI = Comparative Fit Index; TLI = Tucker–Lewis index; SRMR = Standardized Root Mean Square Residual; RMSEA = Root Mean Square Error of Approximation; lower-upper = RMSEA confidence interval’s lower and upper values.

**Table 2 ijerph-20-05827-t002:** Internal consistency analysis.

Reliability		α (95% C.I.)	Ω (95% C.I.)	CR
Burnout	BAT 23	0.94 (0.93–0.94)	0.94 (0.93–0.94)	0.95
	BAT 12	0.88 (0.86–0.90)	0.89 (0.87–0.91)	0.91
Exhaustion	BAT 23	0.85 (0.83–0.87)	0.85 (0.83–0.87)	0.86
	BAT 12	0.77 (0.73–0.81)	0.70 (0.66–0.74)	0.78
Mental Distance	BAT 23	0.79 (0.76–0.82)	0.77 (0.74–0.80)	0.80
	BAT 12	0.67 (0.61–0.73)	0.61 (0.55–0.67)	0.70
Cognitive Impairment	BAT 23	0.81 (0.78–0.84)	0.80 (0.77–0.83)	0.82
	BAT 12	0.83 (0.80–0.86)	0.77 (0.74–0.80)	0.83
Emotional Impairment	BAT 23	0.81 (0.78–0.84)	0.78 (0.75–0.81)	0.81
	BAT 12	0.78 (0.73–0.82)	0.70 (0.65–0.74)	0.77
Psychological complaints	BAT 23	0.75 (0.70–0.79)	0.73 (0.68–0.77)	
Psychosomatic complaints	BAT 23	0.75 (0.71–0.79)	0.71 (0.67–0.75)	

Results of the Cronbach’s α, McDonald’s ω and CR-composite reliability.

**Table 3 ijerph-20-05827-t003:** Average Variance Explained (AVE) and square latent correlations R^2^ for work engagement (UWES), HADS and BAT.

	AVE	R^2^	
		UWES	BAT
UWES	0.726		
BAT Core Symptoms	0.592	0.201	
BAT Secondary Symptoms	0.637	0.017	0.422
		**HADS**	**BAT**
HADS	0.612		
BAT Core Symptoms	0.595	0.386	
BAT Secondary Symptoms	0.658	0.463	0.422

## Data Availability

This research data can be downloaded from GitHub [38].

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
