# Peer review of "The Greek Burnout Assessment Tool: Examining Its Adaptation and Validity"

_ijerph, 2023, doi:10.3390/ijerph20105827_

Round 1

Reviewer 1 Report

Overall, this paper is a well-written and informative contribution to the study of the psychometric properties of the Greek version of the Burnout Assessment Tool (BAT). The authors have provided a clear and concise overview of the BAT. The aims of the study are well-formulated, and the study design and data analysis appear to be rigorous and appropriate. This paper makes an important contribution to the literature by showing that BAT is a useful measurement tool in other cultures.

I think that in the introduction, it would be useful to write a few sentences about the results of the factor analysis of BAT in different countries. In addition to this, it would be useful to write about the practical aspects of the bifactor model in the discussion, e.g., the importance and interpretation of the distinction between main scales and subscales.

Bifactor models are a valuable method of analysis for assessing construct validity in psychological measurement. However, researchers should be cautious when conducting bifactor analysis and avoid using SEM model fit indices as the primary or only criterion for determining the feasibility of a bifactor model, as conclusions based on such criteria can be deceptive given that complementary statistical fit indices provide researchers with pertinent information regarding the salience of general and group-level factors (see, e.g., Rodriguez et al., 2016; Flores-Kanter et al., 2018). Together, this information and the theoretical context will enable the selection of the most appropriate factor model and ensure the validity of inferences drawn from scale scores. It would also be useful to indicate the estimation method used for the CFA.

And when analyzing reliabilities, it would be very useful to calculate model-based reliabilities (hierarchical omega) because it would better show how reliable the subscales are, except for the variance of the group factor (see, e.g., Reise 2012; Rodriguez et al., 2016).

I was pleased to read your well-structured and precisely written manuscript. In my opinion, the additional notes and analysis mentioned above would greatly enhance the value of the paper.

References

Flores-Kanter, P. E., Dominguez-Lara, S., Trógolo, M. A., & Medrano, L. A. (2018). Best practices in the use of bifactor models: conceptual grounds, fit indices and complementary indicators. Revista Evaluar, 18(3).

Reise, S. P. (2012). The rediscovery of bifactor measurement models. Multivariate behavioral research, 47(5), 667-696.

Rodriguez, A., Reise, S. P., & Haviland, M. G. (2016). Evaluating bifactor models: Calculating and interpreting statistical indices. Psychological Methods, 21(2), 137–150.

Author Response

Dear Editor

We would like to thank you and the reviewers for the helpful feedback. We have addressed all the issues raised and changes in the manuscript are indicated via ‘TRACK CHANGES’.

In the following, we will outline how we have addressed the points raises by the reviewers.

Reviewer 1

  1. Comment: I think that in the introduction, it would be useful to write a few sentences about the results of the factor analysis of BAT in different countries.

Our Response: Thank you for your suggestion. In the introduction we have presented the results of the factor analysis of BAT in the different countries accompanied by the appropriate additional references.

  1. Comment: In addition to this, it would be useful to write about the practical aspects of the bifactor model in the discussion, e.g., the importance and interpretation of the distinction between main scales and subscales.

Our Response: Thank you for your suggestion. In the discussion section, the practical aspects of the bifactor model have been noted to highlight the reliability and the precision of the psychometric measurements and interpret the distinctions between the main scales and subscales.

  1. Comment: Bifactor models are a valuable method of analysis for assessing construct validity in psychological measurement. However, researchers should be cautious when conducting bifactor analysis and avoid using SEM model fit indices as the primary or only criterion for determining the feasibility of a bifactor model, as conclusions based on such criteria can be deceptive given that complementary statistical fit indices provide researchers with pertinent information regarding the salience of general and group-level factors (see, e.g., Rodriguez et al., 2016; Flores-Kanter et al., 2018). Together, this information and the theoretical context will enable the selection of the most appropriate factor model and ensure the validity of inferences drawn from scale scores. It would also be useful to indicate the estimation method used for the CFA. And when analyzing reliabilities, it would be very useful to calculate model-based reliabilities (hierarchical omega) because it would better show how reliable the subscales are, except for the variance of the group factor (see, e.g., 2. 2012; Rodriguez et al., 2016).

Our Response: Thank you for your suggestion. The estimation method used for the CFA was the “maximum likelihood” which we have noted in the text. The hierarchical omegas have also been calculated and presented in the results section.

Reviewer 2 Report

Dear Authors,

Congratulations.

I really liked to read your manuscript.

Here are my recommendations:

1-Please do not use abbrivation and write as Burnout Assessment Tool in the title. I think it would be better for the reserachers who search for "burnout" in the title.

2-Please make a clear explanation if you performed analysis of each version on the same sample. If so, please indicate that point as a limitation.

3-Please rethink about the fit indices table and add a coloumn to make more understandable for the reader, and report the good fit or adequate fit based on the "chisquare/degree of freedom" for each model.

4-I recommend avoiding to use a pargraph in quotation marks in the introduction.

Best wishes

I reccommend to get it read by a native English Academic speaker.

Author Response

Dear Editor

We would like to thank you and the reviewers for the helpful feedback. We have addressed all the issues raised and changes in the manuscript are indicated via ‘TRACK CHANGES’.

In the following, we will outline how we have addressed the points raises by the reviewers.

Reviewer 2

  1. Comment: Please do not use abbreviation and write as Burnout Assessment Tool in the title. I think it would be better for the researchers who search for "burnout" in the title.

Our Response: Thank you for your suggestion. In the title, the abbreviation “BAT” has been replaced by “Burnout Assessment Tool”.

  1. Comment: Please make a clear explanation if you performed analysis of each version on the same sample. If so, please indicate that point as a limitation.

Our Response: Thank you for your suggestion. We have noted this as a further limitation emerging from the performance of both versions on the same sample.

  1. Comment: Please rethink about the fit indices table and add a column to make more understandable for the reader, and report the good fit or adequate fit based on the "chi-square/degree of freedom" for each model.

Our Response: Thank you for your suggestion. In the first column of the fit indices (Table 1), we have reported either ‘good fit’ or the ‘adequate fit’ based on the “chi-square/degree of freedom” with the indication (G) or (A) next to the name of each model to indicate the good or adequate fit accordingly.

  1. Comment: recommend avoiding to use a paragraph in quotation marks in the introduction.

Our Response: Thank you for your suggestion. We have removed the cited text and rephrased this section.
